# Impact of Cytoreductive Nephrectomy in the Management of Metastatic Renal Cell Carcinoma: A Multicenter Retrospective Study

**DOI:** 10.3390/diseases12060122

**Published:** 2024-06-04

**Authors:** Naotaka Kumada, Koji Iinuma, Yasuaki Kubota, Kimiaki Takagi, Masahiro Nakano, Takashi Ishida, Shigeaki Yokoi, Fumiya Sugino, Makoto Kawase, Shinichi Takeuchi, Kota Kawase, Daiki Kato, Manabu Takai, Yuki Tobisawa, Takayasu Ito, Keita Nakane, Takuya Koie

**Affiliations:** 1Department of Urology, Graduate School of Medicine, Gifu University, Yanagido 1-1, Gifu 501-1194, Japan or kumamon7821@gmail.com (N.K.); sugino.fumiya.i7@f.gifu-u.ac.jp (F.S.); kawase.makoto.g5@f.gifu-u.ac.jp (M.K.); gallxy7@gifu-u.ac.jp (S.T.); stnf55@gifu-u.ac.jp (K.K.); kato.daiki.m2@f.gifu-u.ac.jp (D.K.); takai.manabu.a5@f.gifu-u.ac.jp (M.T.); tobisawa.yuki.a7@f.gifu-u.ac.jp (Y.T.); nakane.keita@f.gifu-u.ac.jp (K.N.); koie.takuya.h2@f.gifu-u.ac.jp (T.K.); 2Department of Urology, Matsunami General Hospital, 185-1 Kasamatsucho, Hashima-gun, Gifu 501-6062, Japan; 3Department of Urology, Toyota Memorial Hospital, 1-1 Heiwacho, Toyota 471-8513, Japan; yasuaki_kubota@mail.toyota.co.jp; 4Department of Urology, Daiyukai Hospital, 1-9-9 Sakura, Ichinomiya 491-8551, Japan; kimiaki_takagi5619@yahoo.co.jp; 5Department of Urology, Gifu Prefectural General Medical Center, 4-6-1 Noisiki, Gifu 500-8717, Japan; nakano-m@juno.ocn.ne.jp; 6Department of Urology, Gifu Municipal Hospital, 7-1 Kashimacho, Gifu 500-8513, Japan; justaskaxis@gmail.com; 7Department of Urology, Central Japan International Medical Center, Minokamo 505-8510, Japan; s-yokoi@cjimc-hp.jp; 8Center for Clinical Training and Career Development, Graduate School of Medicine, Gifu University, Gifu 501-1194, Japan; ito.takayasu.v9@f.gifu-u.ac.jp

**Keywords:** metastatic renal cell carcinoma, cytoreductive nephrectomy, immune checkpoint inhibitors, targeted therapy, immune checkpoint inhibitors-based combination therapy

## Abstract

In this study, we aimed to determine the utility of cytoreductive nephrectomy (CN) in real-world clinical practice and investigate whether CN contributes to improved oncological outcomes in patients with metastatic renal cell carcinoma (mRCC). This retrospective multicenter cohort study enrolled patients with mRCC who received systemic therapy at six institutions between May 2005 and May 2023. The patients were divided into those who did not undergo CN (Group I) and those who underwent CN (Group II). The primary endpoints were oncological outcomes, including cancer-specific survival (CSS) and progression-free survival (PFS). Altogether, 137 patients with mRCC were included in this study. The median CSS was 14 months in Group I and 32 months in Group II (*p* < 0.001). Additionally, the median PFS in Groups I and II was 5 and 13 months, respectively (*p* = 0.006). A multivariate analysis showed that CN was an independent prognostic factor for CSS and PFS. Hence, CN is a potential treatment modality that can improve oncological outcomes in patients with mRCC.

## 1. Introduction

The incidence rate of renal cell carcinoma (RCC) is 17.1 per 10,000 individuals, whereas the mortality rate is 3.6 per 10,000 individuals [1]. Metastatic RCC (mRCC) accounts for 13% of all RCC cases, with a 5-year relative survival rate of 14% [1]. Although mRCC has a significantly lower incidence rate than localized or regional RCC, it has a worse prognosis [1,2]. The therapeutic landscape for mRCC is currently undergoing rapid evolution with the approval of molecular targeted therapies (MTTs), immune checkpoint inhibitors (ICIs), and combinations of MTTs and ICIs [3,4,5,6,7,8,9].

Nevertheless, the role of cytoreductive nephrectomy (CN) in mRCC remains unclear. CN in patients with mRCC reportedly results in metastatic regression, the prevention of tumor growth, the suppression of new metastases, and a reduction in clinical symptoms by decreasing the release of cytokines and growth factors [10,11,12,13]. In addition, CN is also suggested to reduce paraneoplastic syndromes and improve quality of life (QOL) [14,15]. Two prospective randomized controlled trials in the early 2000s demonstrated that CN with interferon (IFN) significantly improved oncological outcomes compared with IFN alone in patients with mRCC having good performance status [11,16]. Another two prospective randomized trials conducted during the era when MTT was the standard of care for mRCC provided insights into the need for and importance of CN in improving oncological outcomes of patients with mRCC [17,18]. The Clinical Trial to Assess the Importance of Nephrectomy (CARMENA) trial has demonstrated that sunitinib monotherapy was not inferior to CN, with subsequent sunitinib in patients with mRCC who had intermediate or poor prognostic factors according to the Memorial Sloan Kettering Cancer Center (MSKCC) criteria [17]. However, a recent subgroup analysis of the CARMENA trial indicated that overall survival (OS) tended to be longer in patients treated with CN plus sunitinib than in those treated with sunitinib alone; furthermore, patients with mRCC having at least one International Metastatic Renal Cell Carcinoma Database Consortium (IMDC) risk factor may benefit from undergoing upfront CN [19]. The Canadian Uro-Oncology Group jointly conducted the Immediate Surgery or Surgery After Sunitinib Malate in Treating Patients with Metastatic Kidney Cancer (SURTIME) trial, assessing the role of immediate or deferred CN in patients with mRCC who were administered sunitinib, and suggested that the median OS was significantly greater in the deferred CN group than in the immediate CN group [18]. A more recent study reported that the median OS in patients with mRCC was similar between the upfront and deferred CN plus systemic therapy groups, although the upfront CN with systemic therapy group exhibited a significantly better median OS [20]. However, with the combination therapy of TT and ICI as the mainstay of mRCC treatment, there remain numerous unsolved questions regarding the necessity of CN, its timing, and the background of patients who truly need the treatment when the main focus is on improving oncological outcomes.

Various studies have examined the significance of CN in improving the oncological outcomes of patients with mRCC [10,11,12,13,14,15,16,17,18,19]. Notably, a consensus has been reached regarding the significance of CN over the past 20 years [15]. However, significant changes have occurred in the systemic therapy of mRCC, including ICI and MTT, and re-evaluating the significance of CN and the timing of its implementation in current clinical practice may be necessary. Therefore, in this study, we aimed to investigate whether CN contributes to improved oncological outcomes in patients with mRCC and whether the combination of CN and systemic therapy could improve oncological outcomes.

## 2. Materials and Methods

### 2.1. Patient Selection

This retrospective multicenter cohort study included patients with mRCC who received systemic therapy with MTT, ICI, a combination of ICIs, or a combination of MTT and ICI at six institutions between May 2005 and May 2023. The enrolled patients were divided into those who did not undergo CN (Group I) and those who underwent CN (Group II). The following clinical characteristics of the enrolled patients were collected: age, sex, body mass index, Eastern Cooperative Oncology Group performance status (ECOG-PS) [21], primary IMDC risk stratification, tumor–node–metastasis classification according to the American Joint Committee on Cancer (AJCC) staging manual [22], treatment regimen, primary tumor histology, metastatic site, and number of metastases.

This study was approved by the Institutional Review Board of Gifu University (approval number: 2022-125). Owing to the retrospective design of the study, written informed consent was not obtained from the enrolled patients. This is because, in accordance with the Japanese Ethics Committee regulations and ethical guidelines, written consent is not required for retrospective or observational studies using existing or other data in which the results have already been published. The details of this study, which are available only in Japanese, can be found at https://www.med.gifu-u.ac.jp/visitors/disclosure/docs/2022-125.pdf (accessed on 14 August 2023).

### 2.2. Schedule of Systemic Therapy for mRCC

The MTT dose for mRCC was determined based on the strategy used by each treating institution [3,23].

For the ICI combination, 3 mg/kg nivolumab (NIVO) and 1 mg/kg ipilimumab (IPI) were administered intravenously at 3-week intervals to patients with mRCC until September 2018. In October 2018, NIVO was initiated at 240 mg for induction therapy, followed by maintenance therapy every 2 weeks.

For the combination of avelumab (AVE) and axitinib (AXI), AVE was administered intravenously every 2 weeks, whereas AXI was administered orally twice daily [5,24]. Regarding the pembrolizumab (PEM) + AXI regimen, PEM was administered intravenously at a single dose of 200 or 400 mg, whereas AXI was administered orally at 5 mg twice daily [6,13]. When the treatment regimen comprised NIVO and cabozantinib (CABO), NIVO was administered intravenously at a dose of 240 or 480 mg, whereas CABO was administered orally at 40 mg once daily [7,13]. For those receiving PEM and lenvatinib (LEN), PEM was administered intravenously at a dose of 200 or 400 mg, whereas LEN was administered orally at 20 mg once daily [8,13,24]. ICI therapy was discontinued after 2 years in patients treated with ICI + MTT.

Drug treatment method and dosage were determined at each institution, and whether, when, and how to perform CN were also based on the strategy of each institution. Upfront CN referred to CN performed before the initiation of systemic therapy in patients with mRCC who were diagnosed with metastatic disease [13,25]. Deferred CN was defined as a CN performed in patients with mRCC who received systemic therapy. Systemic treatment with or without CN was continued until disease progression was confirmed based on radiological evaluation or the development of tolerable treatment-related adverse events (TRAEs). Radiological progression was defined as the presence of one or more of the following findings: (1) new bone metastases identified on bone scans, computed tomography (CT), or magnetic resonance imaging (MRI); (2) regional soft tissue recurrence or metastasis based on the Response Evaluation Criteria in Solid Tumors guidelines version 1.1 [26]; and (3) at least one new visceral or soft tissue metastasis on CT [2].

### 2.3. Patient Evaluation

Patients were evaluated at baseline before undergoing systemic therapy based on their complete history; physical examination; and chest, abdominal, and pelvic CT and MRI findings. The AJCC staging manual was used to determine the tumor stage [22]. Patient outcomes were evaluated using CT or MRI performed at 1–3-month intervals until radiological disease progression or treatment discontinuation owing to TRAEs.

### 2.4. Statistical Analysis

The primary endpoints were oncological outcomes, including cancer-specific survival (CSS) and progression-free survival (PFS). The JMP 14 software (SAS Institute Inc., Cary, NC, USA) was used for data analysis. The Mann–Whitney U test or Kruskal–Wallis test was used to compare continuous variables between the two groups, whereas the Pearson’s chi-square test or Fisher’s exact test was used to compare changes in each category. The follow-up period was defined as the time from the initial diagnosis of mRCC to the last follow-up examination or date of death. CSS and PFS were determined from the time of initial diagnosis of mRCC to all-cause mortality and disease progression, respectively. Oncological outcomes were analyzed using the Kaplan–Meier method, and log-rank tests were used to examine differences in clinical variables. In the multivariate analysis, the Cox proportional hazards model was used to investigate the clinical factors that influenced oncological outcomes. Statistical significance was defined as a two-sided *p*-value < 0.05.

## 3. Results

### 3.1. Patient Characteristics

Among the 137 patients who were enrolled in this study, 13 were excluded because of selecting the best supportive care (7 patients), missing data (4 patients), receiving CN alone (1 patient), or undergoing renal artery embolization only (one patient). Finally, 124 patients with mRCC were included in the analysis.

Data on pretreatment characteristics of the enrolled patients are presented in Table 1. The patients’ median age was 68 years (interquartile range (IQR), 62–74 years), and the follow-up period was 19 months (IQR, 8–41 months). The follow-up period for patients receiving ICI + ICI, ICI + MTT, MTT alone, and interferon was 17 months (IQR, 8–27.5 months), 9.5 months (IQR, 7–14 months), 21.5 months (IQR, 9.3–55 months), and 49 months (IQR, 28–80 months), respectively. Based on the patients’ history, 57.3% of the patients with mRCC were administered MTT or IFN as the treatment regimen. The male sex was more common than the female sex among the enrolled patients. Additionally, patients in Group I were older and had more cases of clinical ≥ T3 and liver metastases than those in Group II. Only two cases were classified as having a favorable risk; however, this could be because the ECOG-PS relies on the subjective judgment of the primary physician, potentially leading to a possible bias. The median time from surgery to systemic therapy was 1 month (IQR, 0–4 months) in patients who underwent upfront CN, whereas the median time from systemic therapy to CN was 3.5 months (IQR, 1–10 months) in patients who underwent deferred CN. Of the patients classified in Group 2, 18 (25.7%) showed disease progression after the initiation of treatment for mRCC, with a median of 35 months (IQR, 2–5 months). No patient died of any cause during the period between surgery and systemic therapy or between systemic therapy and surgery.

The clinical variables for enrolled patients with good performance status (ECOG-PS ≤ 1) are presented in Table 2.

The types of systemic treatment for patients with mRCC at different time periods are presented. The use of MTT has increased since 2011, followed by a shift to combination therapy, including ICI, after 2019 in patients with mRCC (Figure 1).

### 3.2. Oncological Outcomes

At the end of the follow-up period, 66 (54.8%) patients died from RCC, and 2 (15%) died from other causes (details unknown). The median CSS and PFS of the enrolled patients were 19 (interquartile range (IQR): 8–41) and 8 (IQR: 3–17) months, respectively.

The median CSS was 14 (IQR, 6–27) and 32 (IQR, 18–60) months in Groups I (n = 70) and II (n = 54), respectively (*p* < 0.001; Figure 2a). Additionally, the median PFS in Groups I and II was 5 (IQR, 2–14) and 13 (IQR, 8–28) months, respectively (*p* = 0.006; Figure 2b).

For patients with a good performance status (ECOG-PS ≤ 1), the oncological outcomes are presented in Figure 2. The median CSS was 14 (IQR, 6–29) and 32 (IQR, 17–60) months in Groups I (n = 53) and II (n = 52), respectively (*p* = 0.004; Figure 2a). The median PFS for Groups I and II was 7 (IQR, 3–14) and 13 (IQR, 8–26) months, respectively (*p* = 0.063; Figure 2b).

We investigated the oncological outcomes of patients with mRCC who underwent CN and in those who did not based on their IMDC risk classification. In patients with favorable-intermediate risk, the median CSS was 15 (IQR, 6–29) and 40 (IQR, 25–61) months in Groups I (n = 38) and II (n = 42), respectively (*p* = 0.002; Figure 3a). The median PFS was 14 (IQR, 6–63) and 30 (IQR, 14–87) months in Groups I (n = 38) and II (n = 42), respectively (*p* = 0.061; Figure 3b). 

In patients at poor risk, the median CSS was 12 (IQR, 6–22) and 15 (IQR, 12–22) months in Groups I (n = 32) and II (n = 12), respectively (*p* = 0.262; Figure 4a). The median PFS was 5 (IQR, 3–15) and 48 (IQR, 4–NA) months in Groups I (n = 32) and II (n = 12), respectively (*p* = 0.082; Figure 4b).

We investigated the oncological outcomes of patients with mRCC who received combination therapy including ICI, MTT alone, or IFN as systemic therapy. The median CSS was 32 (IQR, 21-NA) months in patients with mRCC receiving combination therapies including ICI, 79 (IQR, 53–106) months in those receiving MTT alone, and 61 (IQR, 28–81) months in those receiving IFN (Figure 5a). The median PFS was not reached (IQR, 8–NA months) in patients with mRCC who received combination therapies including ICI, 28 (IQR, 12-NA) months for those who received MTT alone, and 30 (IQR, 7–87) months in those who received IFN (Figure 5b). No significant differences in CSS or PFS were observed between the treatment regimens (Figure 6).

The multivariate analysis results indicated that CN was an independent prognostic factor for CSS and PFS (Table 3). Additionally, the examination of a multivariate Cox proportional hazard model with time-dependent covariates showed a hazard ratio of 13.24 (95% confidence interval (CI) 6.59–26.58, *p* < 0.001) for Group I compared to Group II. These results suggest that CN may contribute to the improvement of CSS.

## 4. Discussion

The randomized controlled trial CARMENA demonstrated that sunitinib monotherapy resulted in noninferior OS compared to CN followed by sunitinib in moderate- and poor-risk patients with mRCC, based on the Memorial Sloan Kettering Cancer Center prognostic model [17]. Although the CARMENA study is a prospective study, interpreting the accuracy of the conclusions is difficult because of the following points. First, the study was discontinued because of the inability to accumulate the planned number of patients; second, the enrollment period was 8 years, which is relatively long. Third, only 20% of the centers were able to enroll more than eight cases (more than one case per year); fourth, the number of patients who could be followed up for >2 years was <100 in each group. Lastly, the abovementioned results suggest that the skills of each institution varied considerably, potentially resulting in biases despite the prospective design of the study. Notably, a secondary analysis of the CARMENA trial confirmed that patients who underwent secondary CN had a significantly longer OS than those who did not subsequently undergo CN (median OS: 48.5 and 15.7 months; hazard ratio (HR), 0.34; 95% CI, 0.22–0.54) [19], suggesting that some patients with mRCC benefit from CN treatment. High-intensity CN has been reported to prolong OS in stage IV upper urothelial carcinoma [27], and the usefulness of local therapy for metastatic tumors, as well as mRCC, is of interest in other carcinomas. In the present study, the OS of patients with mRCC who underwent CN was significantly prolonged regardless of the systemic therapy regimen. Furthermore, our results suggest that CN may improve oncological outcomes in patients with mRCC who have good ECOG-PS scores and no other organ metastases, such as brain or liver metastases, which are poor prognostic factors. Previous retrospective analyses of various datasets, including the National Cancer Database and IMDC, have consistently demonstrated that CN confers a survival benefit in patients with mRCCs with clear cell components [28,29]. Therefore, younger patients with mRCC who had fewer IMDC risk factors, no metastases with poor prognosis (such as bone, brain, or liver), and good performance status had a greater potential benefit from CN [13,29].

More than two decades ago, CN was recognized as a potential treatment method to improve oncological outcomes in patients with mRCC, based on the results of randomized controlled trials in which IFN and interleukin were administered as systemic therapy [11,15]. These well-selected patients may benefit from CN in the era of ICI, partly owing to the immune effects of CN [30]. Primary RCC is reportedly associated with the release of cytokines that promote inflammation and inhibit T cells, potentially inhibiting systemic antitumor immune responses [31]. Furthermore, primary renal tumor removal may be beneficial for patients with mRCC eliminating “immune sinks” and reducing the levels of immunosuppressive cytokines, thereby allowing for a more potent antitumor immune response [32]. The benefits of CN can be observed in six categories: regression of metastatic sites; decreased release of cytokines and growth factors that promote metastases; prevention of tumor growth and new metastases; relief of symptoms, such as hematuria and pain; decreased paraneoplastic syndromes; and improved QOL [10,11,12,13,14,15]. In a recent study, a subpopulation of CD133^+^/CD24^+^ cells was identified in clear cell RCC specimens and demonstrated self-renewal and clonogenic pluripotency [33]. Therefore, CN may inhibit tumor growth and metastasis by blocking not only surface proteins but also cancer stem cell signaling-specific pathways. In addition, prolonged OS has been reported in patients with mRCC who underwent multimodal treatment with CN, ICIs, and/or MTTs [13,29,34]. Similarly, CN has been reported to significantly prolong OS in patients with mRCC having various subtypes, such as sarcomatoid differentiation [35]. Thus, nephrectomy may be a potential treatment modality to provide a more effective immune response to mRCC lesions that have been or will be treated with ICIs [13]. Furthermore, the prognosis of patients with mRCC was statistically significantly prolonged after receiving the recent combination therapy including ICI [3,4,5,6,7,8,9]. In this study, we found no significant difference in oncological outcomes because the number of patients who underwent CN after receiving combination therapy including ICI was small and the observation period was short. Future studies in patients with mRCC in accordance with the current treatment strategy are warranted.

One of the main reservations regarding upfront CN for mRCC is the need to consider the risk that 15–30% of patients will not receive subsequent systemic therapy; many of such patients were unable to receive treatment owing to rapid disease progression or postoperative complications [10]. Therefore, a discussion on whether upfront CN should be considered in all patients is necessary.

The first consideration in systemic therapy for mRCC is the establishment of a method to identify patients with biologically aggressive diseases who have a poor prognosis [25]. The phase II study results of 66 patients who received sunitinib before scheduled CN revealed that 27% of the patients had disease progression before or at the time of surgery [36]. In another phase II trial involving 104 patients who received preoperative treatment with pazopanib before planned CN, 16% exhibited disease progression and were unable to receive surgery [37]. Those patients had a shorter OS than those who achieved clinical response (median OS 3.9 months vs. 24.0 months, HR = 3.92; 95% CI 1.78–8.63) [37,38]. Thus, identifying patients who are unlikely to benefit from CN by providing upfront systemic therapy may be possible [25]. However, deferred CN seems to be a treatment worth considering in the era of combination treatment with ICI + MTT, due to the fact that <10% of patients progress after their treatments [6,7,8]. Although some patients may show progression after treatment with ICI + MTT, the establishment of criteria for selecting patients who will benefit from ICI + MTT combination therapy is important.

Controversies remain on whether a combination with modern systemic therapy can improve survival in patients with mRCC and whether CN should be used as an upfront or deferred therapy [38]. A real-world multicenter analysis revealed no correlation between upfront and deferred CN and OS in patients with mRCC who underwent CN and received systemic therapy, although the OS of patients who underwent upfront CN was significantly higher than that of patients who received systemic therapy alone [21]. These results are similar to those of a retrospective study that observed no difference in OS between deferred and upfront CN, although patients with intermediate risk in the MSKCC classification benefited from upfront CN [39]. The SURTIME trial suggested that deferred CN, in which patients initiated systemic treatment with sunitinib and were offered CN if the disease did not progress, might be superior to upfront CN followed by sunitinib therapy for patients with mRCC [18]. In the intention-to-treat population, the 28-week PFS rates were 42% and 43% in the upfront and deferred CN groups, respectively (*p* = 0.61) [18]. At the time of analysis, 70% of patients in the upfront CN group and 57.1% in the deferred CN group had died, with cancer progression being the primary cause in 86% and 89% of patients in the upfront and deferred CN groups, respectively [18]. Additionally, patients receiving sunitinib before CN had a longer median OS than those undergoing upfront CN (32.4 vs. 15.0 months; HR, 0.57; 95% CI, 0.34–0.95) [19]. 

Although the ideal timing for delaying surgery remains unknown, deferred CN may be clinically beneficial in patients who respond to systemic therapy [19]. According to studies using the National Cancer Database, the OS of patients with mRCC who underwent deferred CN was at least equivalent to, if not longer than, that of patients who underwent upfront CN [28,40]. In the present study, although a comparison between upfront CN and deferred CN was not conducted, patients who underwent CN, particularly those with relatively good conditions, such as ECOG-PS ≤ 1 and IMDC favorable-intermediate risk, exhibited better OS and PFS compared with those who did not. Under such conditions, CN may improve oncological outcomes and serve as an effective treatment option. Additionally, considering factors such as QOL and medical costs, performing CN in such patients may be meaningful. Treatment with ICI often allows for the preservation of a relatively good QOL, although MTT therapy may cause challenging side effects, making sustaining treatment difficult and often requiring dose adjustments or interruptions [41]. Therefore, in patients who respond well to systemic therapy, implementing a deferred CN to provide drug-free intervals may improve the QOL. Cases in which deferred CN performed after systemic treatment resulted in no cancer progression, without the need to restart drug therapy, have been documented [42]. While reports on achieving long-term survival after discontinuing drug therapy after deferred CN are still relatively scarce, as similar cases continue to emerge in the future, it may be possible to offer the best treatment for many patients with mRCC. Furthermore, in the treatment of mRCC, the prolonged use of expensive medications poses a significant cost issue for both the patients and society, especially with the increasing prevalence of combination therapies involving ICI + ICI and ICI + MTT [43]. Considering its implication for cost, CN may offer potential improvements. Moreover, despite significant differences in patient backgrounds, choice of regimen in combination with systemic therapy, and timing of CN observed in the multivariate analysis, cytoreductive surgery may have potential benefits in improving oncological outcomes in mRCC patients. In the contemporary era of immunotherapy, multiple ongoing clinical trials are specifically examining the role of CN [44]. If the results of these prospective trials demonstrate the utility of CN, it may lead to the provision of better healthcare in various aspects, such as patients’ oncological outcomes, QOL, and economic costs. Further prospective study results are required to confirm these findings.

This study had some limitations. First, this was a retrospective multicenter, non-randomized study that may have involved a potential inherent bias. In particular, MTTs have been used as the first-line treatment for mRCC for approximately 10 years. In this study, the number of patients treated with combination ICI therapies was smaller than that of patients treated with MTT alone, and oncologic outcomes could not be investigated in detail. Therefore, the true efficacy of ICI + ICI and ICI + MTT in patients with mRCC could not be verified because many of them were treated with MTTs in this study. Second, the number of enrolled patients in this study was relatively small, thereby potentially leading to biased interpretations of the obtained results. The number of patients enrolled in the study was 137, which was very small; therefore, additional cases and long-term follow-up are necessary to determine the accuracy of the results of this study. However, CN has been suggested to be beneficial in some patients in the MTT era. Therefore, a larger number of patients treated with ICI could possibly yield similar results, although the limited sample size remains a limitation. Third, tumor progression, site of metastasis, and patient condition might have a significant impact on the failure to select a CN in patients with mRCC. In this study, a renal tumor biopsy was not performed in most patients in Group I, which may be because of the relatively high number of cases with poor ECOG-PS and the fact that the histological type was predicted based on imaging findings alone rather than based on a biopsy. Fourth, we observed a few cases in which diagnosis was not possible despite performing renal tumor biopsy. Approximately 70% of the patients, mainly in Group I, did not undergo pathological diagnosis by biopsy at the initiation of systemic treatment. In addition to the fact that imaging examinations such as CT and MRI could have predicted the histological type [45], we consider that biopsy was not performed based on the clinical judgment that treatment should be started as soon as possible because of the expected rapid progression of the disease. Empirical therapy might have been administered in many such cases; therefore, the oncological outcomes of Groups I and II could not be accurately evaluated based on the differences in histological types. Fifth, brain metastases were observed only in Group I patients. Liver and bone metastases were observed more frequently in Group I patients than in Group II patients, although without significant difference. Hence, patients in Group I may have additional potentially poor prognostic factors, which may have a negative impact on oncological outcomes. Sixth, this study could not investigate the oncological outcomes according to the timing of surgery (upfront or deferred CN) or the choice of systemic treatment. Upfront CN is generally performed in patients with good performance status and a relatively small number of metastases, whereas deferred CN is limited to patients who respond well to systemic therapy and have good PS at the time of the procedure. Therefore, we cannot eliminate the possibility that this difference influenced the results of the present study. Although the number of patients who underwent CN in this study was significantly smaller than that of patients who did not undergo CN, we evaluated the usefulness of CN in patients who underwent upfront or deferred CN. Seventh, patients who did not receive systemic therapy after CN or those who did not receive CN after systemic therapy because of disease progression or death were not included in this study. An analysis method such as the landmark analysis or time-varying treatment status analysis should have been employed for this study. Finally, the enrollment period of patients in this study was relatively long. Because significant changes in the treatment and surgical procedures for mRCC were observed during this period, the oncological outcome results may have been biased depending on the presence or absence of CN.

## 5. Conclusions

This retrospective multicenter study evaluated the efficacy of CN in patients with mRCC. Patients in Group II had a significantly longer CSS and PFS than those in Group I. Furthermore, the multivariate analysis results revealed that CN was an independent prognostic factor affecting oncological outcomes in patients with mRCC. CN may improve oncological outcomes in patients who receive systemic therapy before and after surgery. Standardizing the systemic treatment regimens and conducting a prospective multicenter study with a short enrollment period may be necessary to examine the efficacy of CN for mRCC.

## Figures and Tables

**Figure 1 diseases-12-00122-f001:**
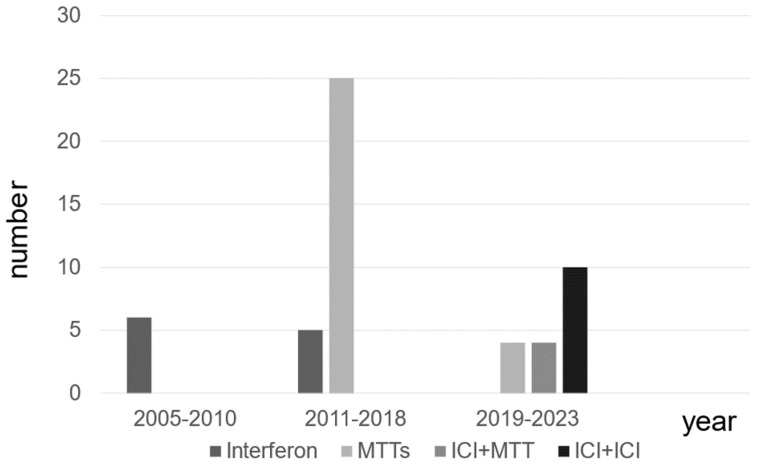
The distribution of systemic therapies for patients with metastatic renal cell carcinoma.

**Figure 2 diseases-12-00122-f002:**
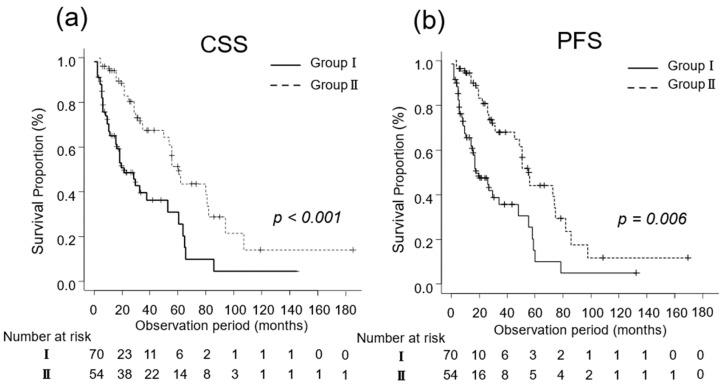
Kaplan–Meier estimates of cancer-specific survival (CSS) and progression-free survival (PFS) in patients with metastatic renal cell carcinoma who did not undergo cytoreductive nephrectomy (Group I) and in those who underwent cytoreductive nephrectomy (Group II). (**a**) The median CSS is 14 months in Group I and 32 months in Group II (*p* < 0.001). (**b**) The 3-year PFS rate is 30.4% and 42.8% in Groups I and II, respectively (*p* = 0.006).

**Figure 3 diseases-12-00122-f003:**
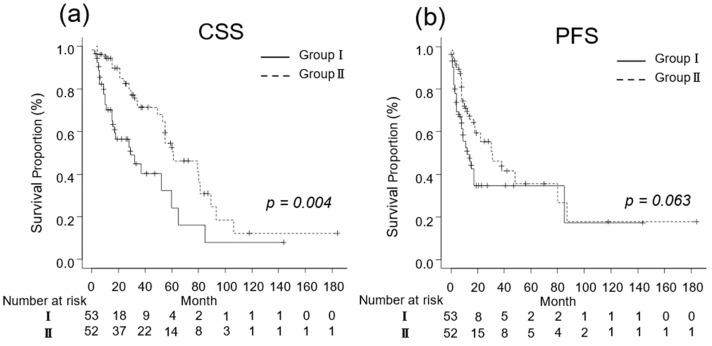
Kaplan–Meier estimates of cancer-specific survival (CSS) and progression-free survival (PFS) in metastatic renal cell carcinoma patients with Eastern Cooperative Oncology Group performance status ≤ 1 who did not undergo cytoreductive nephrectomy (Group I) and in those who underwent cytoreductive nephrectomy (Group II). (**a**) The median CSS is 14 and 32 months in Groups I and II, respectively (*p* = 0.004; Figure 2a). (**b**) The 3-year PFS rate is 34.8% and 46.2% in Groups I and II, respectively (*p* = 0.063).

**Figure 4 diseases-12-00122-f004:**
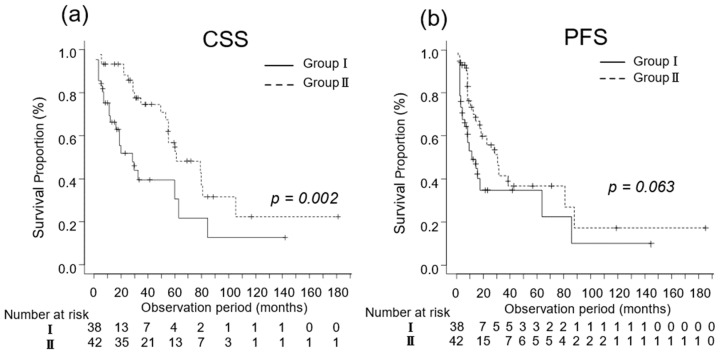
Kaplan–Meier estimates of cancer-specific survival (CSS) and progression-free survival (PFS) in patients with favorable-intermediate risk metastatic renal cell carcinoma who did not undergo cytoreductive nephrectomy (Group I) and in those who underwent cytoreductive nephrectomy (Group II) based on their International Metastatic Renal Cell Carcinoma Database Consortium risk classification. (**a**) The median CSS is 15 and 40 months in Groups I and II, respectively (*p* = 0.002; Figure 3a). (**b**) The 3-year PFS rate is 36.2% in Group I and 42.7% in Group II (*p* = 0.061).

**Figure 5 diseases-12-00122-f005:**
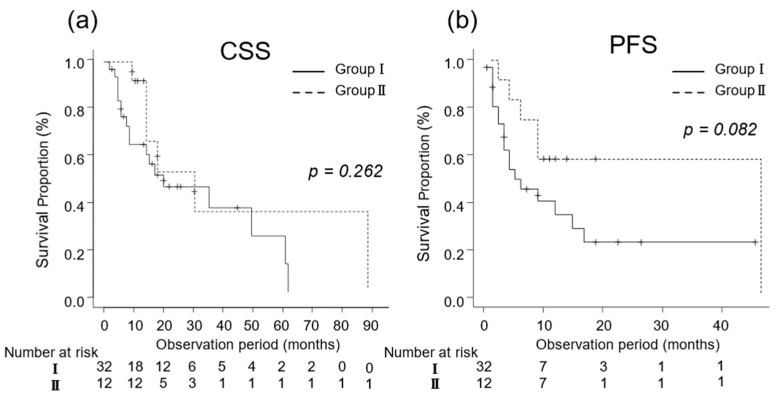
Kaplan–Meier estimates of cancer-specific survival (CSS) and progression-free survival (PFS) in patients with poor risk metastatic renal cell carcinoma who did not undergo cytoreductive nephrectomy (Group I) and in those who underwent cytoreductive nephrectomy (Group II) based on their International Metastatic Renal Cell Carcinoma Database Consortium risk classification. (**a**) The median CSS is 12 and 15 months in Groups I and II, respectively (*p* = 0.262; Figure 4a). (**b**) The 3-year PFS rate is 23.1% in Group I and 58.3% in Group II (*p* = 0.082).

**Figure 6 diseases-12-00122-f006:**
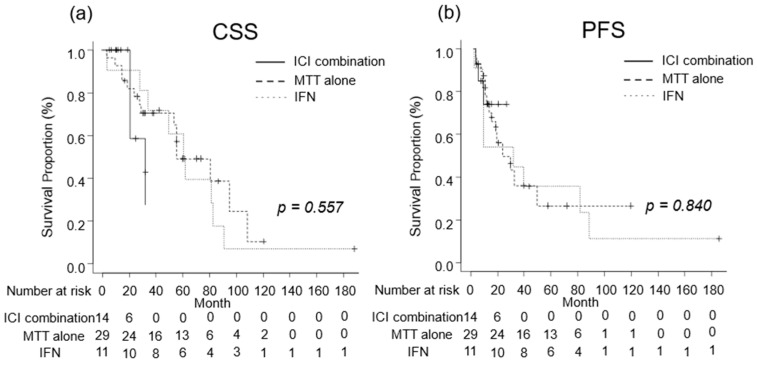
Oncological outcomes by treatment regimen were evaluated using the Kaplan–Meier curves. (**a**) The median cancer-specific survival was 32 months in patients treated with combination therapy including immune checkpoint inhibitor (ICI), 79 months in those treated with molecular targeted therapy (MTT) alone, and 61 months in those treated with Interferon (IFN). (**b**) The median progression-free survival in patients who received combination therapy including ICI, MTT alone, and IFN alone were not reached, 28 months, and 30 months, respectively.

**Table 1 diseases-12-00122-t001:** Patient characteristics.

Covariates	Group I	Group II	*p*-Value
Patients (number)	70	54	
Age (year, median, IQR)	69 (63–76)	66 (61–72)	0.062
Sex (number, %)			0.067
Male	56 (80.0)	35 (64.8)	
Female	14 (20.0)	19 (35.2)	
ECOG-PS			<0.001
≤1	53 (75.8)	52 (96.2)	
≥2	17 (24.2)	2 (3.8)	
Primary IMDC risk classification (number, %)			0.004
Favorable	2 (2.9)	0 (0.0)	
Intermediate	36 (51.4)	42 (77.8)	
Poor	32 (45.7)	12 (22.2)	
Clinical T stage (number, %)			0.062
T1	15 (21.4)	14 (25.9)	
T2	14 (20.0)	19 (35.2)	
T3	24 (34.3)	17 (31.5)	
T4	16 (22.9)	4 (7.4)	
Tx	1 (1.4)	0 (0.0)	
Clinical N status (number, %)			0.307
N0	38 (54.3)	31 (57.4)	
N1	26 (37.1)	22 (40.7)	
Nx	6 (8.6)	1 (1.9)	
Types of systemic therapy			0.163
ICI + ICI	21 (30.0)	10 (18.5)	
ICI + MTT	8 (11.4)	4 (7.4)	
MTT alone	35 (50.0)	29 (53.7)	
Interferon	6 (8.6)	11 (20.4)	
Type of pathology			<0.001
Clear cell	16 (22.9)	41 (75.9)	
Non-clear cell	2 (2.9)	9 (16.7)	
Unknown	52 (74.3)	4 (7.4)	
Metastatic sites			
Lung	49 (70.0)	36 (66.7)	0.701
Brain	9 (12.9)	0 (0.0)	0.005
Liver	17 (20.7)	3 (5.5)	0.056
Bone	32 (45.7)	17 (31.5)	0.139
Number of metastases			0.006
1	17 (24.3)	28 (51.8)	
≥2	53 (75.7)	26 (48.2)	

Group I, patients with metastatic renal cell carcinoma who did not undergo cytoreductive nephrectomy; Group II, patients with metastatic renal cell carcinoma who underwent cytoreductive nephrectomy; IQR, interquartile range; ECOG-PS, Eastern Cooperative Oncology Group performance status; IMDC, International Metastatic Renal Cell Carcinoma Database Consortium; ICI, immune checkpoint inhibitor; MTT, molecular targeted therapy.

**Table 2 diseases-12-00122-t002:** Demographic data of enrolled patients with good performance status.

	Group I	Group II	*p*-Value
Patients (number)	53	52	
Age (year, median, IQR)	70 (65–77)	66 (61–72)	0.009
Sex (number, %)			0.012
Male	46 (86.8)	34 (65.4)	
Female	7 (13.2)	18 (34.6)	
Primary IMDC risk classification (number, %)			0.044
Favorable	1 (1.9)	0 (0.0)	
Intermediate	31 (58.5)	41 (78.8)	
Poor	21 (39.6)	11 (21.2)	
Clinical T stage (number, %)			0.007
T1	12 (22.6)	13 (25.0)	
T2	9 (17.0)	19 (36.5)	
T3	16 (30.2)	17 (32.7)	
T4	15 (28.3)	3 (5.8)	
Tx	1 (1.9)	0 (0.0)	
Clinical N status (number, %)			0.326
N0	29 (54.7)	30 (57.7)	
N1	19 (35.8)	21 (40.4)	
Nx	5 (9.4)	1 (1.9)	
Types of systemic therapy			0.137
ICI + ICI	17 (32.1)	10 (19.2)	
ICI + MTT	6 (11.3)	4 (7.7)	
MTT alone	26 (49.1)	27 (51.9)	
Interferon	4 (7.5)	11 (21.2)	
Type of pathology			<0.001
Clear cell	13 (24.5)	39 (75.0)	
Non-clear cell	2 (3.8)	9 (17.3)	
Unknown	38 (71.7)	4 (7.7)	
Metastatic sites			
Lung	37 (69.8)	35 (67.3)	0.835
Brain	6 (11.3)	0 (0.0)	0.027
Liver	9 (17.0)	3 (5.8)	0.123
Bone	19 (35.8)	16 (30.8)	0.680
Number of metastases			0.038
1	15 (28.3)	27 (51.9)	
≥2	38 (71.7)	25 (48.1)	

Group I, patients with metastatic renal cell carcinoma who did not undergo cytoreductive nephrectomy; Group II, patients with metastatic renal cell carcinoma who underwent cytoreductive nephrectomy; IQR, interquartile range; ECOG-PS, Eastern Cooperative Oncology Group performance status; IMDC, International Metastatic Renal Cell Carcinoma Database Consortium; ICI, immune checkpoint inhibitor; MTT, molecular targeted therapy.

**Table 3 diseases-12-00122-t003:** Multivariate analysis according to oncological outcomes.

Variables	Number	Cancer-Specific Survival			Progression-Free Survival		
		HR	*p*-Value	95% CI	HR	*p*-Value	95% CI
Age							
<70	61	1 (ref.)			1 (ref.)		
≥70	44	1.015	0.306	0.986–1.045	1.009	0.554	0.980–1.039
Sex							
Male	80	1 (ref.)			1 (ref.)		
Female	25	0.349	0.007	0.162–1.045	0.401	0.013	0.195–0.823
ECOG-PS							
0	73	1 (ref.)			1 (ref.)		
1	32	2.168	0.034	1.060–4.435	1.074	0.847	0.517–2.232
Clininal T stage							
≤2	53	1 (ref.)			1 (ref.)		
≥3	52	1.791	0.001	1.289–2.489	1.810	0.001	1.289–2.540
Liver metastasis							
Negative	93	1 (ref.)			1 (ref.)		
Positive	12	1.759	0.214	0.722–4.288	2.846	0.027	1.125–7.197
Brain metastasis							
Negative	99	1 (ref.)			1 (ref.)		
Positive	6	6.283	0.005	1.733–22.780	9.429	0.001	2.514–35.360
Type of pathology							
Clear cell	52	1 (ref.)			1 (ref.)		
Non-clear cell/unknown	53	1.401	<0.001	1.160–1.692	1.327	0.001	1.122–1.569
Cytoreductive nephrectomy							
Performed	52	1 (ref.)			1 (ref.)		
Not performed	53	2.618	0.044	1.027–6.675	2.671	0.028	0.110–6.423

HR, hazard ratio; CI, confidence interval; ref., reference; IMDC, International Metastatic Renal Cell Carcinoma Database Consortium.

## Data Availability

Data and material are provided in this paper.

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
