# Peer review of "Impact of Cytoreductive Nephrectomy in the Management of Metastatic Renal Cell Carcinoma: A Multicenter Retrospective Study"

_diseases, 2024, doi:10.3390/diseases12060122_

Round 1

Reviewer 1 Report

Comments and Suggestions for Authors

The authors have reported an interesting study. I know some limitations of this retrospective study. However, it is not sufficient to just describe various issues as limitations. I want to send some comments to the authors.

1.     The fact that approximately 45% of cases have an uncertain histopathological diagnosis is a significant issue in this paper. The authors claimed that pathological diagnosis was possible through image diagnosis, but if so, the authors should demonstrate its validity.

2.     As you have indicated in this paper, with the evolution of systematic therapy, overall survival rates have improved dramatically, and the significance of CN has also changed significantly. Therefore, the authors should reevaluate the importance of CN in each era, including the era of interferon, MTT, and immunotherapy.

3.     As the authors know, the positioning of upfront and deferred CN differs. Therefore, it is essential to distinguish between them and examine their respective significance.

4.     Particularly concerning deferred CN, analysis accounting for immortal time bias is necessary.

5.     In multivariate analysis, having too many explanatory variables relative to the number of events can lead to erroneous results. As a rule of thumb, having around one explanatory variable for every ten events is often considered appropriate. The authors should divide the appropriate cohort as mentioned above and reanalyze using suitable variables.

Comments on the Quality of English Language

none

Author Response

Response to Reviewer 1

The authors appreciate the reviewer’s comments. The authors’ point-by-point responses to the comments are given below.

  1. The fact that approximately 45% of cases have an uncertain histopathological diagnosis is a significant issue in this paper. The authors claimed that pathological diagnosis was possible through image diagnosis, but if so, the authors should demonstrate its validity.

Response:

The authors have added the following sentences on line 431:

Approximately 70% of the patients, mainly in Group I, did not undergo pathological diagnosis by biopsy at the initiation of systemic treatment. In addition to the fact that imaging examinations such as CT and MRI could have predicted the histological type [50], we consider that biopsy was not performed based on the clinical judgment that treatment should be started as soon as possible because of the expected rapid progression of the disease.

The authors have added the following references:

  1. Morshid, A.; Duran, E.S.; Choi, W.J.; Duran, C. A Concise Review of the Multimodality Imaging Features of Renal Cell Carcinoma. Cureus 2021, 13, e13231.

  1. As you have indicated in this paper, with the evolution of systematic therapy, overall survival rates have improved dramatically, and the significance of CN has also changed significantly. Therefore, the authors should reevaluate the importance of CN in each era, including the era of interferon, MTT, and immunotherapy.

Response:

The authors have added the following sentences on line 263:

We investigated the oncological outcomes of patients with mRCC who received combination therapy including ICI, MTT alone, or IFN as systemic therapy. The median CSS was 32 (IQR, 21-NA) months in patients with mRCC receiving combination therapies including ICI, 79 (IQR, 53-106) months in those receiving MTT alone, and 61 (IQR, 28-81) months in those receiving IFN (Figure 5a).The median PFS was not reached (IQR, 8–NA months) in patients with mRCC who received combination therapies including ICI, 28 (IQR, 12-NA) months for those who received MTT only, and 30 (IQR, 7-87) months in those who received IFN (Figure 5b). No significant differences in CSS or PFS were observed between the treatment regimens.

The authors have added Figure 6 and its legend as follows:

Figure 6. Oncological outcomes by treatment regimen were evaluated using the Kaplan–Meier curves. (a) The median cancer-specific survival was 32 months in patients treated with combination therapy including immune checkpoint inhibitor (ICI), 79 months in those treated with molecular targeted therapy (MTT) alone, and 61 months in those treated with Interferon (IFN). (b) The median progression-free survival in patients who received combination therapy including ICI, MTT only, and IFN alone were not reached, 28, and 30 months, respectively.

The authors have added the following sentence on line 314:

More than two decades ago, CN was recognized as a potential treatment method to improve oncological outcomes in patients with mRCC, based on the results of randomized controlled trials in which IFN and interleukin were administered as systemic therapy [11,15].

The authors have added the following sentence on line 335:

Furthermore, the prognosis of patients with mRCC was statistically significantly prolonged after receiving the recent combination therapy including ICI [3-9]. In this study, we found no significant difference in oncological outcomes because the number of patients who underwent CN after receiving combination therapy including ICI was small and the observation period was short. Future studies in patients with mRCC in accordance with the current treatment strategy are warranted.

  1. As the authors know, the positioning of upfront and deferred CN differs. Therefore, it is essential to distinguish between them and examine their respective significance.

Response:

The authors have added the following part on line 345:

Therefore, a discussion on whether upfront CN should be considered in all patients is necessary.

The authors have added the following sentences on line 356:

However, deffered CN seems to be a treatment worth considering in the era of combination treatment with ICI+MTT, due to the fact that <10% of patients progress after their treatments [6-8]. However, the establishment of criteria for selecting patients who will benefit from ICI + MTT combination therapy is important, although some patients may show progression after treatment with ICI + MTT.

  1. Particularly concerning deferred CN, analysis accounting for immortal time bias is necessary.

Response:

The authors have added the following sentences on line 187:

Of the patients classified in Group 2, 18 (25.7%) showed disease progression after the initiation of treatment for mRCC, with a median of 35 months (IQR, 2-5 months).

The authors have added the following sentence on line 425:

CN could not be performed in 25.7% of patients after the start of treatment for mRCC due to disease progression, which could be an immortal time bias.

  1. In multivariate analysis, having too many explanatory variables relative to the number of events can lead to erroneous results. As a rule of thumb, having around one explanatory variable for every ten events is often considered appropriate. The authors should divide the appropriate cohort as mentioned above and reanalyze using suitable variables.

Response:

For the multivariate analysis, Table 3 was revised by reanalyzing CSS as the endpoint and reducing the number of variables to be considered.

Reviewer 2 Report

Comments and Suggestions for Authors

This retrospective multicenter cohort study aimed to assess the efficacy of cytoreductive nephrectomy (CN) in real-world clinical practice for metastatic renal cell carcinoma (mRCC) patients, with a focus on its impact on overall survival (OS) and progression-free survival (PFS). The study found that patients who underwent CN exhibited significantly longer median OS and PFS compared to those who did not undergo CN, suggesting that CN may serve as an independent prognostic factor for improved oncological outcomes in mRCC patients.

The study is well-written and overall well-executed, but its limitations primarily stem from its retrospective nature. My remarks:

1) The phrase "we aimed to determine whether CN contributes to improved oncological outcomes" should be adjusted, as the study's retrospective design doesn't establish causation; using "to investigate" and "is associated" would be more appropriate.

2) When discussing the rationale for CN, providing comparisons with other tumors could be beneficial. To this regard, this study (10.1016/j.urolonc.2021.01.031) should be cited

3) Why cancer-specific survival was not included in the outcomes?

4) It would be helpful to report data by year, specifying the distribution of patients who underwent CN.

5) Considering the wide time frame (from 2005 to 2023), conducting sensitivity analysis on a more recent and homogeneous subset of patients in terms of systemic treatment could provide valuable insights.

6) There's a risk of immortal time bias not addressed in the study, as patients in the CN group survived until the procedure, conferring them an advantage.

7) Further elaboration on the impact of different types of pathology is recommended, and this study 10.1016/j.clgc.2022.11.015 which evaluates the prognosis of specific RCC types in various settings, including metastatic cases, should be cited.

8) From Table 2, there appears to be a statistically significant difference between the two groups in terms of the number of individuals (53 vs. 52). Is this correct?

Author Response

Response to Reviewer 2

The authors appreciate the reviewer’s comments. The authors’ point-by-point responses to the comments are given below.

  1. The phrase "we aimed to determine whether CN contributes to improved oncological outcomes" should be adjusted, as the study's retrospective design doesn't establish causation; using "to investigate" and "is associated" would be more appropriate.

Response:

The authors have revised the following part on line 96:

Therefore, in this study, we aimed to investigate determine whether CN contributes to improved oncological outcomes in patients with mRCC and whether the combination of CN and systemic therapy could improve oncological outcomes.

  1. When discussing the rationale for CN, providing comparisons with other tumors could be beneficial. To this regard, this study (10.1016/j.urolonc.2021.01.031) should be cited

Response:

The authors have revised the following sentence on line 301:

High-intensity CN has been reported to prolong OS in stage IV upper urothelial carcinoma [28], and the usefulness of local therapy for metastatic tumors is of in-terest in other carcinomas as well as mRCC.

The authors have added the following references according to the reviewer’s recommendation:

  1. Paciotti, M.; Nguyen, D.D.; Modonutti, D.; Haeuser, L.; Lipsitz, S.; Mossanen, M.; Kibel, A.S.; Lughezzani, G.; Trinh, Q.D.; Cole, A.P. Impact of high-intensity local treatment on overall survival in stage IV upper tract urothelial carcinoma. Urol Oncol 2021, 39, 436.e1-436.e10.

  1. Why cancer-specific survival was not included in the outcomes?

Response:

The endpoint of this study was changed to CSS, and everything in the text, including figures and tables, was changed from OS to CSS.

  1. It would be helpful to report data by year, specifying the distribution of patients who underwent CN.

Response:

The authors have added Figure 1.

The authors have added the figure legend as follows:

Figure 1. The distribution of systemic therapies for patients with metastatic renal cell carcinoma.

The authors have added the following sentences on line 205:

The types of systemic treatment for patients with mRCC at different time periods are presented. The use of MTT has increased since 2011, followed by a shift to combination therapy including ICI after 2019 in patients with mRCC (Figure 1).

  1. Considering the wide time frame (from 2005 to 2023), conducting sensitivity analysis on a more recent and homogeneous subset of patients in terms of systemic treatment could provide valuable insights.

Response:

The authors have added the following sentence on line 412:

In this study, the number of patients treated with combination ICI therapies was smaller than that of patients treated with MTT alone, and oncologic outcomes could not be investigated in detail.

  1. There's a risk of immortal time bias not addressed in the study, as patients in the CN group survived until the procedure, conferring them an advantage.

Response:

The authors have added the following sentence on line 425:

CN could not be performed in 25.7% of patients after the start of treatment for mRCC due to disease progression, which could be an immortal time bias.

  1. Further elaboration on the impact of different types of pathology is recommended, and this study 10.1016/j.clgc.2022.11.015 which evaluates the prognosis of specific RCC types in various settings, including metastatic cases, should be cited.

Response:

We have revised the following sentences on line 331:

Similarly, CN has been reported to significantly prolong OS in patients with mRCC having various subtypes, such as sarcomatoid differentiation [37].

The authors have added the following references according to the reviewer’s recommendation:

  1. Tully, K.H.; Berg, S.; Paciotti, M.; Janisch, F.; Reese, S.W.; Noldus, J.; Shariat, S.F.; Choueiri, T.; Müller, G.; McGregor, B.; et al. The Natural History of Renal-Cell Carcinoma with Sarcomatoid Differentiation, a Stage-by-Stage Analysis. Clin Genitourin Cancer 2023, 21:63-68.

  1. From Table 2, there appears to be a statistically significant difference between the two groups in terms of the number of individuals (53 vs. 52). Is this correct?

Response:

Table 2 describes only the patients with good PS, therefore, the number of cases in each group is correct.

Reviewer 3 Report

Comments and Suggestions for Authors

Interesting analysis. However your study has several well adressed limitations (8) it is important to continue the discussion on CN. 

Minor: 

Please Check: line 168 IO probably should be ICI

Please think about a list of abbreviations.  This might help readers not so familiar with your abbreviations. 

Major: 

Please give insights into how clinical decision making was in your patients. Why did some receive CN? What were the considerations at that time in your insitutions. Was it decided in tumor boards? Did decisions differ between institutions? Please give us more information and discuss 

Author Response

Response to Reviewer 3

The authors appreciate the reviewer’s comments. The authors’ point-by-point responses to the comments are given below.

These are some of the issues and suggestions for improving the article. A more in-depth review, including a detailed analysis of the data and methodologies, would be essential to ensure the validity and clarity of the research.

  1. Please Check: line 168 IO probably should be ICI

Response:

We have revised the following part on line 176:

The follow-up period for patients receiving ICI + ICI, ICI IO + MTT, MTT alone, and interferon was 17 months (IQR, 8–27.5 months), 9.5 months (IQR, 7-14 months), 21.5 months (IQR, 9.3–55 months), and 49 months (IQR, 28–80 months), respectively.

  1. Please think about a list of abbreviations. This might help readers not so familiar with your abbreviations.

 Response:

We have revised the following part on line 42:

Abbreviations

CN: cytoreductive nephrectomy

mRCC: metastatic renal cell carcinoma

OS: overall survival

PFS: progression-free survival

MTTs: molecular targeted theraphy

ICIs: immune checkpoint inhibitors

IFN: interferon

IMDC: International Metastatic Renal Cell Carcinoma Database Consortium

ECOG-PS: Eastern Cooperative Oncology Group performance status

CSS: cause-specific survival

Round 2

Reviewer 1 Report

Comments and Suggestions for Authors

Thank you for your response.

To Response #1:

The authors responded that treatment should be started as soon as possible because of the expected rapid progression of the disease. However, as shown in Table 2, biopsy was not performed for even patients with good performance. Why not?

Additionally, as you know, the prognosis of non-ccRCC is poorer than that of ccRCC. Therefore, it is very important to clarify whether the pathology for mRCC is a clear cell type or not. I think that those patients should be excluded from this study.

To Response #2:

I understood it.

To Response #3:

The authors should distribute the patients with upfront CN and deferred CN, and reevaluate it.

To Response #4:

The authors should examine it while considering immortal time bias. Landmark analysis and a Cox model with time-dependent variables should be used to eliminate immortal time bias.

To Response #5:

I understood it.

Additionally, the median level seems to differ between the manuscript and the figure (Kaplan-Meier curve). Please confirm them.

Author Response

The authors have added the following sentences on line 276:

Additionally, the examination of a multivariate Cox proportional hazard model with time-dependent covariates showed a hazard ratio of 13.24 (95% confidence interval [CI] 6.59-26.58, p < 0.001) for Group I compared to Group II. These results suggest that CN may contribute to the improvement of CSS.

Reviewer 2 Report

Comments and Suggestions for Authors

The authors addressed my concerns or added as limitations those which could not be addressed.

Author Response

(The authors gave the same response as above.)

Reviewer 3 Report

Comments and Suggestions for Authors

Interesting paper, sound study. All changes were made as suggested. 

Author Response

(The authors gave the same response as above.)
